# Global trends in antimicrobial use in food-producing animals: 2020 to 2030

**Ranya Mulchandani**[1], **Yu Wang**[1], **Marius Gilbert**[2,3], **Thomas P. Van Boeckel**[1,4]*

**1** Health Geography and Policy Group, ETH Zürich, Zurich, Switzerland, **2** Spatial Epidemiology Lab, Université Libre de Bruxelles, Brussels, Belgium, **3** Fonds National de la Recherche Scientifique, Brussels, Belgium, **4** Center for Diseases Dynamics, Economics, and Policy, New Delhi, India

* thomas.van.boeckel@gmail.com

**Data Availability Statement:** The dataset used for this analysis - which included data extracted from published literature - is within the Supporting Information files.

## Abstract

Use of antimicrobials in farming has enabled the growth of intensive animal production and helped in meeting the global increase in demand for animal protein. However, the widespread use of veterinary antimicrobials drives antimicrobial resistance, with important consequences for animal health, and potentially human health. Global monitoring of antimicrobial use is essential: first, to track progress in reducing the reliance of farming on antimicrobials. Second, to identify countries where antimicrobial-stewardship efforts should be targeted to curb antimicrobial resistance. Data on usage of antimicrobials in food animals were collected from 42 countries. Multivariate regression models were used in combination with projections of animal counts for cattle, sheep, chicken, and pigs from the Food and Agriculture Organization to estimate global antimicrobial usage of veterinary antimicrobials in 2020 and 2030. Maps of animal densities were used to identify geographic hotspots of antimicrobial use. In each country, estimates of antimicrobial use (tonnes) were calibrated to match continental-level reports of antimicrobial use intensity (milligrams per kilogram of animal) from the World Organization for Animal Health, as well as country-level reports of antimicrobial use from countries that made this information publicly available. Globally, antimicrobial usage was estimated at 99,502 tonnes (95% CI 68,535–198,052) in 2020 and is projected, based on current trends, to increase by 8.0% to 107,472 tonnes (95% CI: 75,927–202,661) by 2030. Hotspots of antimicrobial use were overwhelmingly in Asia (67%), while <1% were in Africa. Findings indicate higher global antimicrobial usage in 2030 compared to prior projections that used data from 2017; this is likely associated with an upward revision of antimicrobial use in Asia/Oceania (~6,000 tonnes) and the Americas (~4,000 tonnes). National-level reporting of antimicrobial use should be encouraged to better evaluate the impact of national policies on antimicrobial use levels.

## Introduction

Demand for animal protein has been rising globally over the past decade [1]. Meeting this growing demand has been facilitated by the expansion of intensive animal production systems where antimicrobials are used routinely to maintain health and productivity [2]. In 2017,

**Funding:** RM was supported by the European Union's Horizon 2020 grant for MOOD (Monitoring Outbreaks for Disease surveillance in a data science context) (No 874850). MG was supported by the Belgian Fonds National de la Recherche Scientifique. TPVB and YW were supported by The Swiss National Science Foundation Eccellenza Fellowship (No 181248), and the Branco Weiss Foundation. The funders played no role in the design or interpretation of the study.

**Competing interests:** The authors have declared no competing interests exist.

antimicrobial use (AMU) in animals represented 73% of all antimicrobials used worldwide [3], and its use contributes to the rise of antimicrobial resistance (AMR) [4]. In animals, AMR can result in treatment failure [5], and thus represents a threat to the long-term sustainability of the animal industry. In humans, drug-resistant infections resulting from veterinary antimicrobial use remains challenging to quantify [6, 7] but may, for certain drug-pathogens combinations, pose a serious threat to human health [8, 9].

Monitoring global AMU is essential for tracking progress in addressing the causes of AMR. However, current initiatives are heterogenous across regions. In Europe, the harmonized Surveillance of Veterinary Consumption (ESVAC) report has been in place since 2005, and currently reports data from 31 countries in the European Union (EU) [10]. Outside of the EU, only 9 countries routinely publish national reports on AMU (S1 Table).

Since 2016, the World Organization for Animal Health (WOAH, formerly the Office International des Epizooties) annually gathers data, on a voluntary basis, from up to 157 countries on their use of antimicrobial agents in animals [11]. However, prior to public release, the country-level data collected as part of these annual surveys is aggregated in five regions (Europe, Africa, Americas, Middle East, and Asia/Far East/Oceania). Furthermore, the list of countries that contributes data on AMU within each region is not publicly available. As such, this unidentifiability of countries which report to WOAH, and the regional aggregation of national AMU reports, precludes establishing a truly global AMU monitoring system. Firstly, it prevents evaluating the effect of national policies and stewardship efforts on AMU levels. Secondly, the acknowledgment of countries that have successfully reduced AMU might be diluted at regional level if neighbouring countries compensate with an increase in AMU. Thirdly, it prevents further epidemiological analyses, such as evaluating a country's antibiotic footprint using national trade data, which would also benefit from AMU at a national level.

In the absence of national-level data, previous studies have utilised modelling approaches to extrapolate for non-reporting countries using usage data reported by countries for 2010, 2015 and 2017 [2, 3, 12]. However, AMU regulation and policy is a dynamic landscape. In 2021, for the first time, the population-weighted mean antimicrobial use (AMU) in food-producing animals was lower than in humans in the EU/EEA [13]. This reduction of AMU in food-producing animals could be explained by policies and stewardship efforts [14, 15]. For example, in Nordic countries antimicrobials can only be obtained by veterinarian's prescription, who should follow guidelines on antimicrobial treatments of animals (when to treat, dosage, administration route etc.), and who are not allowed to make a profit from their sales [16], decentivising (over)use of antimicrobials. However, policies governing AMU in animal production in other continents still vary widely between countries [17]. Brazil—the largest meat exporter in the world—still largely lacks a legal framework on the use of antimicrobials [18]. In contrast, other major meat producers such as China have recently implemented strategies for reducing AMU and in 2017 banned colistin as a feed additive [19]. Therefore, global estimates of antimicrobial use in animals need to be regularly revised, and new trends must be interpreted in light of this ever-changing regulatory landscape.

In this study, statistical models [2] were used in combination with country reports of veterinary AMU, regional totals of AMU from WOAH, and on-farm AMU surveys, to estimate AMU in food-producing animals for 229 countries/territories in 2020. Global AMU was estimated to 2030, and AMU was mapped at 10x10km resolution to identify global hotspots of high AMU.

## Materials and methods

### Ethics statement

No ethical approval was required for this study. All data were retrieved from open access.

## Antimicrobial usage data

An online search was conducted from three sources of data 1) government reports on veterinary antimicrobials (predominantly sales) at country-level, collected through established surveillance systems, 2) scientific articles reporting estimates of veterinary AMU at country-level (predominantly imports), and 3) scientific articles reporting on-farm AMU from surveys within countries. Articles that contained data on antimicrobial sales, consumption, usage, or imports were all considered as a proxy for usage (henceforth referred to as usage), in line with WOAH reporting criteria on AMU [11]. Four groups of animals were included: cattle, sheep, chicken, and pigs, which total 91.1% of animal biomass raised for food globally [20].

The literature search was conducted in PubMed using the following search terms: Veterinary (antibiotic OR antimicrobial) (use OR usage OR consumption OR sales) AND (antibiotic OR antimicrobial) (use OR usage OR consumption OR sales) AND (animals OR food animals OR livestock OR swine* OR pig* OR poultry* OR chicken* OR cattle* OR dairy* OR beef* OR sheep*). Between 01 January 2010 and 01 June 2022, 4,217 articles were identified, and all had their titles and abstracts screened. 126 articles underwent a full text read. 10 articles contained national data on antimicrobial sales, import, or consumption. Of these, four articles came from countries that did not have any national surveillance system; however, only half (n = 2) contained data post-2017 and only those were retained (Cameroon, Timor-Leste) (S2 Table). An additional four articles were identified to contain sub-national species-level survey data only (Brazil, Fiji, Morocco, and Pakistan) (S3 Table).

Combining data from government reports and scientific articles, AMU estimates pooled across animal species were identified for 42 countries that reported data after 2017 (the reference year for the last global-level analysis of national sales data [12]) (S1 and S2 Tables). Breakdown of antimicrobial sales by compounds were pooled in 13 classes: Tetracyclines, Amphenicols, Penicillins, Cephalosporins, Sulfonamides, Macrolides, Lincosamides, Aminoglycosides, Quinolones, Pleuromutilins, Polymyxins, and other/not-reported. Quinolones included sub-class fluoroquinolones, and sulphonamides included sub-class trimethoprims. In addition, species-specific estimates of AMU after 2010 were identified in 19 countries, five of which did not report AMU pooled across species. Sub-categories of animal species such as broilers and layer chickens were pooled together.

The dataset included all countries contributing to ESVAC. For European countries that reported both in ESVAC and in national reports, pooled data (for all animal species) from ESVAC was preferred to facilitate comparability between countries. As species-level data is not currently reported to ESVAC, this data was extracted from the national reports when available (S3 Table). The final dataset did not include countries that had published estimates of AMU older than the year 2018. The motivation for this modelling choice was to prevent biasing global estimates of AMU with outdated papers/reports that may not reflect ongoing efforts to curb AMU. These included: Australia (2010), Iran (2010), Pakistan (2017), Singapore (2017), South Africa (2015), Tanzania (2017), and Viet Nam (2015). Of these, reports (rather than on-farm surveys) were identified from Australia, Singapore, and South Africa, however these were one off exercises rather than related to any new routine surveillance and no subsequent reports have been published since.

## Food animal census

The total biomass of animals was estimated in each country or region or territory (henceforth referred to as "country") using population correction units (PCU). The PCU represents the total number of animals in a country (alive or slaughtered), multiplied by the average weight of the animal at the time of treatment. Therefore, the PCU is a standardization metric that

accounts for differences in animal weight, and number of production cycles per year between countries. PCUs were calculated with data from FAOSTAT for 2020 [21] and Gilbert et al., 2018 [23], using:

$$PCU_{k,s} = An_{k,s} \cdot (1 + n_{k,s}) \cdot \left(\frac{Y_k}{R_{\frac{CW}{LW},k}}\right)$$

where $An_{k,s}$ is the number of animal type, $k$, for each production system, $s$ (either intensive or extensive) in each country. $n_{k,s}$ is the number of production cycles for each animal type in each production system. $Y_k$ is the quantity of meat in each country for each animal type. $R_{CW/LW,k}$ is the carcass weight to live weight ratio for each animal type [22]. Data for pigs and chickens was disaggregated by production system according to Gilbert et al., 2018 [23].

### Extrapolation of antimicrobial usage in non-reporting countries

A five-step statistical procedure was used to estimate overall AMU in 187 countries from 42 countries that reported sales, consumption, or import data of veterinary antimicrobials, using a baseline year of 2020. This statistical procedure was initially developed by Van Boeckel et al., 2015 [2], and adapted in Tiseo et al., 2021 [12]. In this study, the methodology was further updated such that country estimates of AMU were calibrated to match continent-level reports of AMU intensity (mg/kg) from WOAH (S1 Protocol). All statistical analyses were done in R version 4.1.0.

### Projection between 2020 and 2030

AMU was projected in 229 countries between 2020 and 2030 under current trends for AMU intensity [mg/PCU] per animal species, antibiotic classes, and production system. The stock of cattle, sheep, chicken, and pigs was projected in each country in 2030 using the BAU scenario of the "*The Future of Food and Agriculture*: *Alternative Pathways to 2050*" maintained by FAO [24]. Concretely, stocks obtained from FAOSTAT for 2020 were multiplied by the ratio of projection for the stock of animals in the BAU between 2020 and 2030. For chicken and pigs, the proportion of animals raised in extensive production systems in 2030 was extrapolated using projections of GDP/capita from International Monetary Fund (IMF) [25] following the relationship proposed by Gilbert et al., 2015 [26]. The projections of GDP per capita of the IMF were extended from 2027 to 2030 using linear regression models.

### Mapping

AMU was mapped at 10x10 kilometres resolution using global layers of animal densities [23]. The coefficient of AMU intensity, $\alpha_{c,k,s}$ [mg/PCU], obtained for each country, class of antibiotic and animal species (S1 Protocol) were multiplied by the PCU in each pixel. The PCU in each pixel was calculated as the product of the stock of animals of each species in each pixel with productivity factor obtained at the national level, $n_{k,s}$, $Y_k$, and $R_{\frac{CW}{LW},k}$ (See '*Food animal census*'). Maps of cattle and sheep [23], and chickens and pigs [26], were obtained from the 4[th] iteration of the Gridded Livestock of the World (GLW) for the year 2015 [27]. Pigs raised in intensive and semi-intensive systems were pooled into the same production system whereas pigs raised in extensive systems were mapped separately. All maps of animal densities were scaled such that the sum of animals of each pixel inside a country matched the stocks of animals reported by each country to FAOSTAT in 2020 [21]. Maps of animal densities were projected to 2030 using the methodology described in '*Projection from 2020 to 2030*' applied on each 10x10 kilometre pixel.

## Results

### Global trends in antimicrobial use

In 2020, global AMU (tonnes) for cattle, sheep, chicken, and pigs was estimated at 99,502 tonnes of active ingredient (95% CI: 68,535–193,052). Based on current trends for AMU intensity (mg/PCU), global AMU was projected to increase by 8.0% to 107,472 tonnes (95% CI: 75,927–202,661) by 2030. Global AMU intensity was also projected to increase from 79.3 mg/PCU in 2020 to 85.6 mg/PCU by 2030. AMU intensity varied considerably between countries in 2020, ranging from 337.8 mg/PCU in Thailand to 4.4 mg/PCU in Norway (S2 Fig). In 2020, usage per biomass unit for sheep was 243.3 mg/PCU, pigs was 173.1 mg/PCU, cattle was 59.6 mg/PCU, and chicken was 35.4 mg/PCU.

### Projected antimicrobial usage by country in 2020 and 2030

The top 5 consumers in 2020 were China, Brazil, India, USA, and Australia (Figs 1A and 2). Together these countries made up 58% of global AMU; they were also predicted to remain the top 5 in 2030. Pakistan was predicted to have the largest relative increase in AMU (44%) (from 2,184 to 3,143 tonnes), followed by Australia (16%). Asia consumed the majority of antimicrobials in 2020 (58,377 tonnes, 59%), including 32,776 tonnes (56%) from China alone (Fig 1A and 1C). Africa, Oceania, and South America, despite lower absolute AMU, were predicted higher relative increases of AMU by 25%, 16%, and 14% respectively.

### Trends in antimicrobial use by antimicrobial class

Tetracyclines were the most commonly used antimicrobial overall (33,305 tonnes) (S3 Fig) and were predicted to increase by 9% by 2030 (Fig 1B). However, AMU intensity per antimicrobial class varied by country. For example, Thailand had the highest proportion for penicillins, while Chile had the highest proportion for amphenicols (S2 Fig).

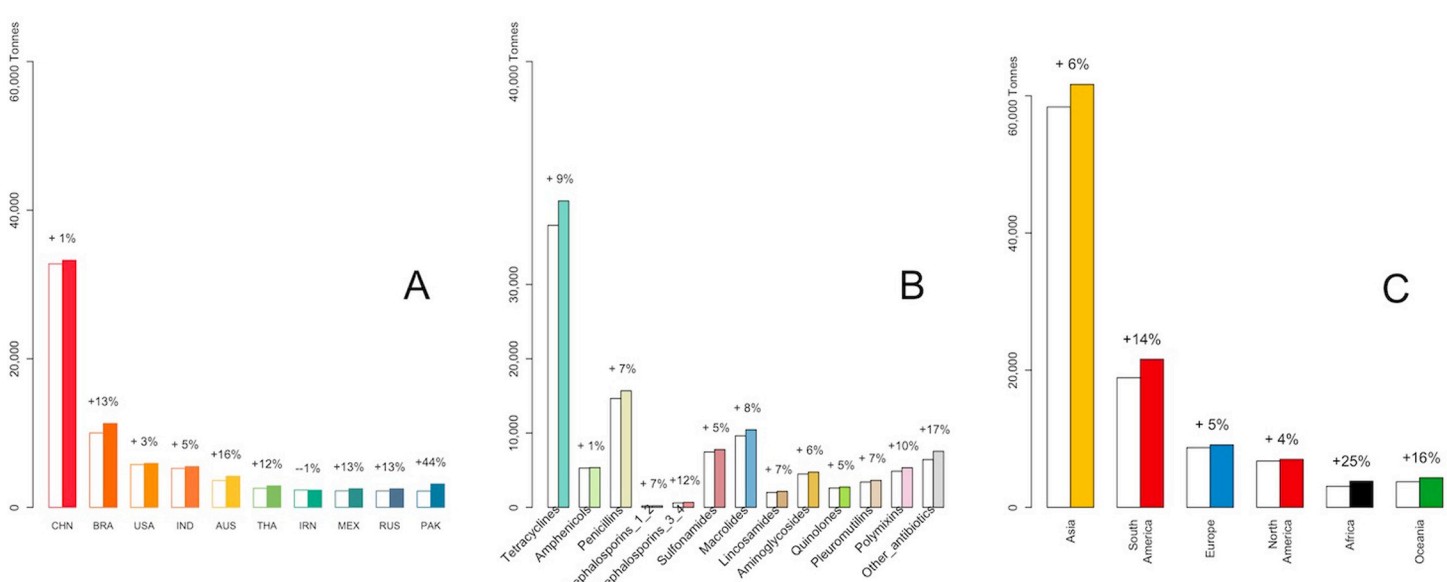

**Fig 1.** Veterinary antimicrobial consumption in 2020 (white bars) and their projected consumption for 2030 (coloured bars) by **(A)** country (top 10), **(B)** antimicrobial class, and **(C)** continent. CHN, China; BRA, Brazil; IND, India; USA, United States; AUS, Australia; IRN, Iran; THA, Thailand; PAK, Pakistan; JPN, Japan; MEX, Mexico.

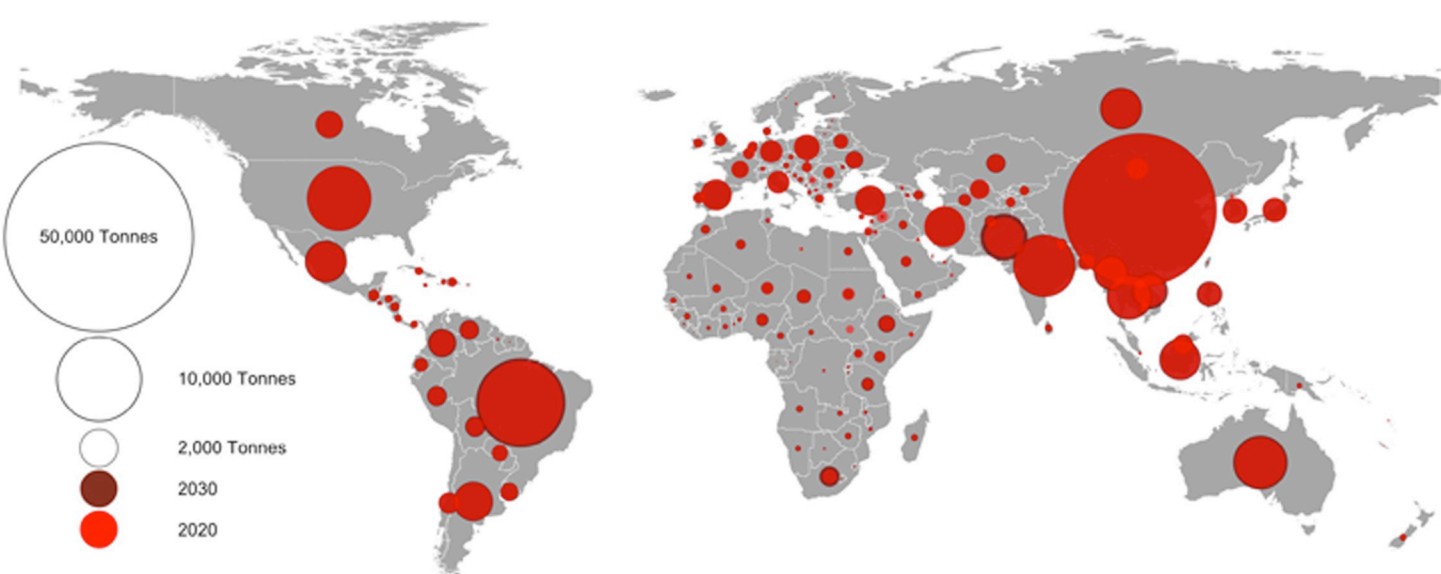

**Fig 2. Antimicrobial consumption per country in 2020 and 2030.** Circles are proportional to quantity of antimicrobials used. Red circles correspond to the quantity used in 2020, and outer dark red ring corresponds to the projected increase in consumption in 2030. Country boundaries were obtained from GADM (https://gadm.org/download_world40.html).

### Geographical hotspots of antimicrobial use

The majority of hotspot areas of AMU intensity were found in Asia (67%). These were identified in eastern China, southern India, Central Java (Indonesia), central Thailand, eastern coastline of Viet Nam, western South Korea, eastern India and Bangladesh, Pakistan, and northwest Iran. In Oceania, the main hotspot was observed along the south-eastern coast of Australia. Within Europe, hotspots were identified in northern Italy, northern Germany, and central Poland. In the Americas, most hotspot areas were identified in the south of Brazil and in the Midwest of the USA. Less than 1% of AMU intensity hotspots were identified in Africa, and these were confined to the Nile delta, and peri-urban areas of Johannesburg (Fig 3).

## Discussion

In this study, we estimated the global usage of veterinary antimicrobials in 2020 at 99,502 tonnes. In comparison, in 2018, WOAH reported a lower global AMU estimate of 76,704 tonnes across 109 countries that responded to their annual survey. WOAH collects a subjective estimate of coverage within each country and uses this to adjust country estimates upwards; however, it does not make any extrapolations for countries that do not respond–which in 2018 was 40% of countries worldwide. In this study, the analysis extrapolated AMU for 229 countries. AMU in Africa was predicted to be double that reported by WOAH (3,065 vs 1,477 tonnes, respectively) and AMU in the region Asia/Far East/Oceania was 50% higher than estimates from WOAH (61,883 vs 44,621 tonnes, respectively). These regional differences may be explained by the proportion of countries that responded to the WOAH survey per region and therefore contributed to their AMU totals: which was 44% (n = 24) of countries in Africa and 68% (n = 22) in Asia/Far East/Oceania.

In the most recent WOAH report, a decline in global AMU was reported between 2016–2018 (from 92,269 to 69,455 tonnes); this is in fact largely attributable to a decline in AMU in China (44,186 to 29,774 tonnes), as reported by the Chinese Ministry of Agriculture [28]. However, these efforts seem to have plateaued between 2018–2020, as reported in the present analysis and

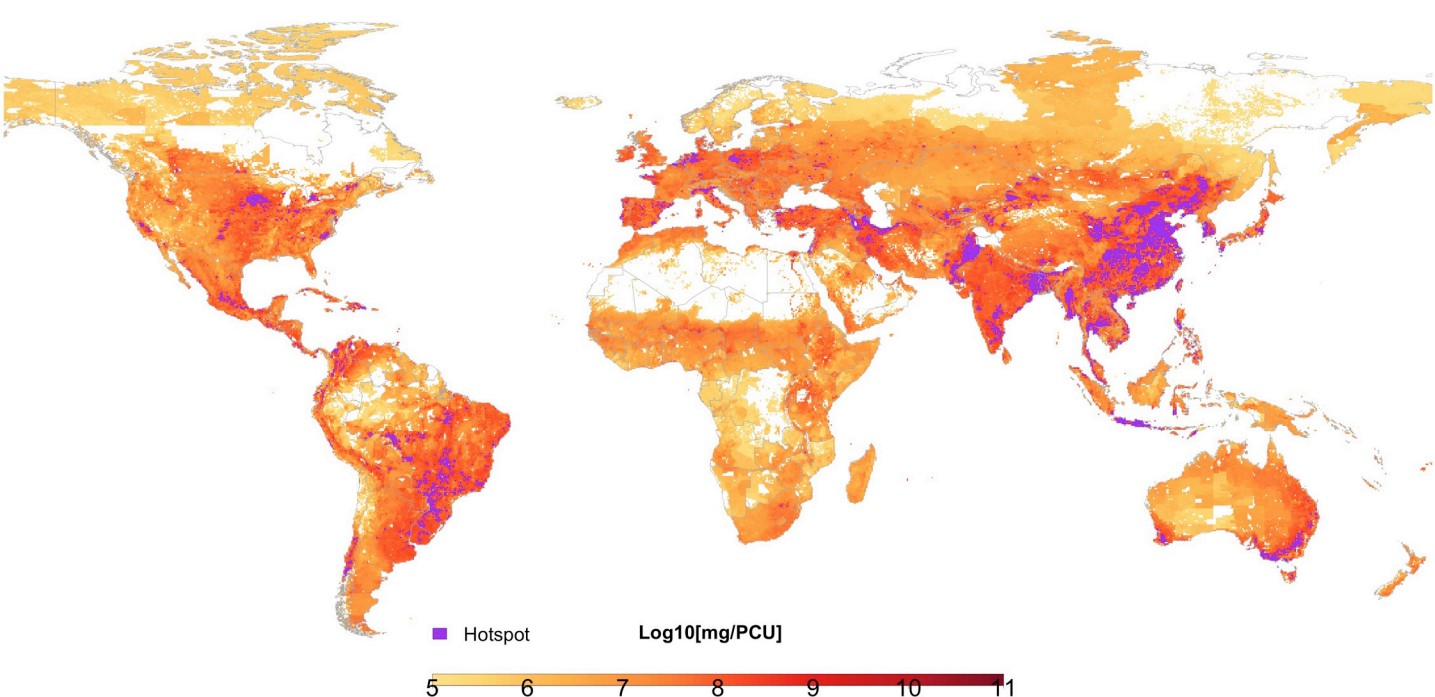

**Fig 3. Global distribution of veterinary antimicrobial consumption at 10 x 10 kilometers resolution expressed in milligrams per biomass (population correction units).** Purple indicates hotspot areas (top 95% percentile). Country boundaries were obtained from GADM (https://gadm.org/download_world40.html).

Chinese reports (30,904 and 32,763 tonnes in 2019 and 2020, respectively). This reduction may therefore not be observed in future rounds of WOAH reporting when using a 2020 baseline and may explain the discrepancies in overall yearly trends of AMU at a global level.

No additional countries have commenced national reporting of AMU since the previous analysis for 2017 [12]. As such, in this analysis, regional totals were matched to WOAH AMU intensity levels to allow for the predictions to include data from a greater number of countries (109 rather than just 42 with national reports), albeit in an aggregated form. This could explain the difference in the revised estimates of global AMU for 2030 (107,472 tonnes), which were slightly higher than previous predictions [12] of 104,079 tonnes by 2030.

In 2020, China, Brazil, India, and United States remain in the top 5 countries for absolute AMU in tonnes, in line with previous predictions for 2017 and 2010 [2, 12]. In contrast, Australia now also appears among the top 5 –however, in the absence of any public report of AMU since 2010 [29], usage for Australia was obtained through extrapolation, as with other countries that do not report data. Therefore, predictions may not accurately reflect the current country efforts to reduce AMU.

Despite national actions plans to tackle AMR and selected bans on antimicrobials in animal production [30] there is still no publicly available country-level reports of veterinary AMU in the majority of countries of the world. In particular 6 out of the 10 largest meat producers in the world do not report AMU to the public (Brazil, Russia, Mexico, Argentina, India, Viet Nam) [21]. Countries that export a significant share of their animal production have been previously shown to be more likely to report AMU data [12]; again, Brazil is a notable exception, as despite being the largest exporter of poultry and cattle in the world [20] it still does not openly publish its AMU data.

Estimates of AMU intensity in 2020 for pigs remained comparable to that for 2017 (173 vs 193 mg/PCU), while chicken halved again (148 mg/PCU for 2010, 68 mg/PCU for 2017, 35

mg/PCU for 2020), despite the continued growth of the poultry production market [1]. This could be explained by numerous husbandry factors that have been applied extensively throughout the chicken industry. For example, countries may have switched from medically important antibiotics to the use of ionophores which are not included in AMU data published by most countries as they are not considered veterinary medicines. In the UK between 2013 and 2017, despite the use of medically important antibiotics declining from 94 tonnes to 14 tonnes, its use of ionophores went from 209 tonnes to 281 tonnes during the same period, keeping its overall AMU relatively stable [31]. Another explanation could be changes in chicken breeds. In the Netherlands, AMU reduction in chickens could be explained by their large-scale transition from fast-growing to slow-growing chicken breeds, which have been shown to require fewer antibiotic treatments due to their improved overall welfare [32]. Conversely, in cattle, despite similar predictions for 2010 and 2017 (45 mg/PCU and 42 mg/PCU respectively), AMU increased in 2020 to 60 mg/PCU [2, 12]. This trend is potentially attributable to the upward revision of AMU in the Americas and Asia, resulting from matching regional AMU intensity reported by WOAH. In particular, now major cattle producing countries such as Brazil, Pakistan, India, and Australia are included, which had not previously been in the global analyses to quantify AMU.

As with any modelling study, this analysis comes with limitations. Firstly, the number of countries for which country-level AMU reports could be identified between 2018 and 2020 was limited (n = 42), the majority of which came from high-income countries. Since the first assessment of global AMU was carried out in 2015 [2] the number of countries that reports AMU publicly has only grown marginally from 32 to 42. Therefore, our methodology was adapted to incorporate data collected by WOAH from 109 countries to supplement the analysis. However, the major limitation of these data was that they were pooled by region prior to release, and effectively only provided 5 additional data points to inform the present analysis. As shown in the case of China, national level would be beneficial to better understand trends, as their decline followed by subsequent plateau in AMU from 2016–2019 was not observable in global or regional data.

Secondly, the analysis was restricted to four species: cattle, sheep, chicken, and pigs. For this, the 4th GLW was utilised, a resource available on livestock densities worldwide. However, in particular for sheep and pigs, there are some data gaps which limit the ability to model AMU in these areas (e.g., sheep in Vietnam). Additionally, although these four species make up >90% of animal biomass raised for food, less abundant species have been associated with very high AMU intensity (e.g., 390 mg/kg in rabbits in France [33]). Additional animal species should be integrated in future global AMU assessments. However, these are currently only reported in a handful of countries (i.e., n = 2 for rabbits). Furthermore, the present analysis does not include aquaculture, which was recently quantified by Schar and colleagues in 2020 [34] in an independent study, however it also suffers from a lack of data precluding meaningful updates.

Thirdly, projections within this study were conducted under a BAU scenario. As such, where countries take actions to curb AMU in the coming years, these projections with be likely over-estimates of AMU by 2030.

Since 2015, an increasing number of countries have started reporting national AMU data, and the availability of data on global AMU has gradually improved. However, a majority of countries still do not publish these data publicly. Availability of national data would enable comparisons within regions that have large discrepancies between countries, and perhaps more importantly, relate antibiotic stewardship policies (or absence thereof) with accurate AMU levels. Until these data are publicly available, for any national level analysis of the impact of AMU we continue to rely on modelling estimates. Therefore, member states of WOAH can

play an essential role to support global efforts in tackling AMR by empowering it with the ability to report country-level estimates of AMU.

## Supporting information

**S1 Protocol. Extrapolation of antimicrobial usage.**
(DOCX)

**S1 Fig. Share of tetracycline in a country vs antimicrobial usage per kilogram of animal (AMU intensity, mg/PCU).** Circles are proportional to the log10 of the PCU in each country.
(TIFF)

**S2 Fig. Veterinary antimicrobial usage intensity by country in 2020, split by antimicrobial class.** All countries are referred to by their ISO3 alpha-3 code, found: https://www.iso.org/obp/ui/#search.
(TIFF)

**S3 Fig.** Proportion of veterinary antimicrobial usage in 2020 by (A) antimicrobial class, (B) animal species, and (C) type of animal.
(TIFF)

**S4 Fig. Veterinary antimicrobial usage intensity by antimicrobial class, split by animal species in 2020.**
(TIFF)

**S5 Fig.** Global distribution of veterinary antimicrobial usage at 10 x 10 km resolution expressed in milligram per biomass (population correction units) for (A) pooled animal species (B) cattle (C) chicken (D) pigs and (E) sheep. Purple indicates hotspot areas (top 95 percentile). Country boundaries were obtained from GADM (https://gadm.org/download_world40.html).
(TIFF)

**S1 Table. National antimicrobial usage reports (overall) with data since 2018 (n = 40).**
(XLSX)

**S2 Table. Point-prevalence surveys on antimicrobial imports in countries with an absence of national AMU surveillance (n = 2).**
(XLSX)

**S3 Table. National species-specific AMU reports and on-farm usage surveys (\*) with data since 2010 (n = 19).** \* intensity (mg/kg) was used rather than tonnage due to sub-national coverage of survey.
(XLSX)

**S1 Data. This file contains all the data extracted from the published literature, as outlined in the methods section of the main manuscript.**
(CSV)

## Author Contributions

**Conceptualization:** Thomas P. Van Boeckel.

**Data curation:** Ranya Mulchandani, Yu Wang, Marius Gilbert.

**Formal analysis:** Thomas P. Van Boeckel.

**Funding acquisition:** Thomas P. Van Boeckel.

**Investigation:** Thomas P. Van Boeckel.

**Methodology:** Thomas P. Van Boeckel.

**Supervision:** Thomas P. Van Boeckel.

**Visualization:** Thomas P. Van Boeckel.

**Writing – original draft:** Ranya Mulchandani.

**Writing – review & editing:** Thomas P. Van Boeckel.

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
