## [Decision Letter · Decision Letter 0]

6 Oct 2022

PGPH-D-22-01542

Global trends in antimicrobial use in food-producing animals: 2020 to 2030

Dear Dr. Mulchandani,

Thank you for submitting your manuscript to PLOS Global Public Health. After careful consideration, we feel that it has merit but does not fully meet PLOS Global Public Health’s publication criteria as it currently stands. Therefore, we invite you to submit a revised version of the manuscript that addresses the points raised during the review process. not, for example, on novelty or perceived impact.

We look forward to receiving your revised manuscript.

Kind regards,

Ismail Ayoade Odetokun, DVM, Ph.D.

Academic Editor

Journal Requirements:

1. Please review your reference list to ensure that it is complete and correct. If you have cited papers that have been retracted, please include the rationale for doing so in the manuscript text, or remove these references and replace them with relevant current referen1.ces. Any changes to the reference list should be mentioned in the rebuttal letter that accompanies your revised manuscript. If you need to cite a retracted article, indicate the article’s retracted status in the References list and also include a citation and full reference for the retraction notice.

2. Please send a completed 'Competing Interests' statement, including any COIs declared by your co-authors. If you have no competing interests to declare, please state "The authors have declared that no competing interests exist". Otherwise please declare all competing interests beginning with the statement "I have read the journal's policy and the authors of this manuscript have the following competing interests:"

3. Please insert an Ethics Statement at the beginning of your Methods section, under a subheading 'Ethics Statement'. It must include:

a) The name(s) of the Institutional Review Board(s) or Ethics Committee(s)

b) The approval number(s), or a statement that approval was granted by the named board(s) 

c) (for human participants/donors) - A statement that formal consent was obtained (must state whether verbal/written) OR the reason consent was not obtained (e.g. anonymity). NOTE: If child participants, the statement must declare that formal consent was obtained from the parent/guardian.

4. Please provide separate figure files in .tif or .eps format only and remove any figures embedded in your manuscript file. Please also ensure that all files are under our size limit of 10MB.

Additional Editor Comments (if provided):

Please, kindly respond to the issues raised within the manuscript my the reviewers.

Reviewers' comments:

Reviewer's Responses to Questions

**Comments to the Author**

1. Does this manuscript meet PLOS Global Public Health’s publication criteria? Is the manuscript technically sound, and do the data support the conclusions? The manuscript must describe methodologically and ethically rigorous research with conclusions that are appropriately drawn based on the data presented.

Reviewer #1: Yes

Reviewer #2: Yes

Reviewer #3: Yes

2. Has the statistical analysis been performed appropriately and rigorously?

Reviewer #1: Yes

Reviewer #2: Yes

Reviewer #3: Yes

3. Have the authors made all data underlying the findings in their manuscript fully available (please refer to the Data Availability Statement at the start of the manuscript PDF file)?

Reviewer #1: Yes

Reviewer #2: Yes

Reviewer #3: Yes

4. Is the manuscript presented in an intelligible fashion and written in standard English?

Reviewer #1: Yes

Reviewer #2: Yes

Reviewer #3: Yes

5. Review Comments to the Author

Reviewer #1: Here the authors present the Global trends in antimicrobial use in food-producing animals from 2020 to 2030. Antimicrobial resistance is a major public health and clinical concern globally, and antimicrobial use in food animals have been implicated as one of its drivers. The concept of the study is timely, as the objectives provide additional information. The concept, study design, and analysis of the study were very good, including the presentation of results and discussion.

Some minor comments to improve the quality of the manuscript.

Line 28: Add "in" after "helped".

Line 36: Change "animals" to "animal"

Line 129: Delete "cephalosporins" as it appears twice.

Line 130: Correct spelling of "Polymixins" to " Polymyxins". Add "and" after "Polymyxins" and "s" to "other". Delete ")" after "other".

Line 132: Write "5" in words.

Line 136: Complete sentence "for pooled...across animal species" what? Not clear

Line 145: Correct "on-off" to "one off".

Reviewer #2: An interesting manuscript on the Global trends in antimicrobial use in food-producing animals: 2020 to 2030. In this study, the authors estimated antimicrobial use (tonnes) calibrated to match continental-level reports of antimicrobial use intensity (milligrams per kilogram of animal) from the World Organization of Animal Health, as well as country-level reports of antimicrobial use from countries. Very good efforts.

The manuscript is well articulated but needs minor review.

L101: Insert ‘of’ after ‘…. number’.

L250: Insert ‘comma ( , )’after ‘… tonnes’

L251: Insert ‘comma ( , )’after ‘… tonnes’

Reviewer #3: This manuscript is technically sound and does have verifiable data to support findings and conclusions, but not without minor errors that will require revision.

Information on lines 97 through 103 is better suited for materials and methods, while emphasis should be placed on the paper's stated goal.

Line 107 should be restructured for clarity. The report of "veterinary usage" stated in the sentence was unclear.

Line 114 through 124 should be restructured to include the year range for literature search, which were not stated in the manuscript or the supporting document.

The authors need to provide information on how bias/risk assessments were carried out in the materials and methods,

The authors did not state clearly or highlight their conclusions and recommendations that could negate any possible negative projections made.

In conclusion, authors are advised to subject this write-up to the assessment of a native English speaker to take care of minor typographical and grammar errors.

6. PLOS authors have the option to publish the peer review history of their article (what does this mean?). If published, this will include your full peer review and any attached files.

**Do you want your identity to be public for this peer review?** For information about this choice, including consent withdrawal, please see our Privacy Policy.

Reviewer #1: No

Reviewer #2: No

Reviewer #3: **Yes: **Oluwadamilola Abiodun-Adewusi

---

## [Decision Letter · Decision Letter 1]

5 Dec 2022

Global trends in antimicrobial use in food-producing animals: 2020 to 2030

PGPH-D-22-01542R1

Dear Ms Mulchandani,

We are pleased to inform you that your manuscript 'Global trends in antimicrobial use in food-producing animals: 2020 to 2030' has been provisionally accepted for publication in PLOS Global Public Health.

Best regards,

Ismail Ayoade Odetokun, DVM, Ph.D.

Academic Editor

Reviewer Comments (if any, and for reference):

Reviewer's Responses to Questions

**Comments to the Author**

1. If the authors have adequately addressed your comments raised in a previous round of review and you feel that this manuscript is now acceptable for publication, you may indicate that here to bypass the “Comments to the Author” section, enter your conflict of interest statement in the “Confidential to Editor” section, and submit your "Accept" recommendation.

Reviewer #1: All comments have been addressed

Reviewer #2: All comments have been addressed

2. Does this manuscript meet PLOS Global Public Health’s publication criteria? Is the manuscript technically sound, and do the data support the conclusions? The manuscript must describe methodologically and ethically rigorous research with conclusions that are appropriately drawn based on the data presented.

Reviewer #1: Yes

Reviewer #2: Yes

3. Has the statistical analysis been performed appropriately and rigorously?

Reviewer #1: Yes

Reviewer #2: Yes

4. Have the authors made all data underlying the findings in their manuscript fully available (please refer to the Data Availability Statement at the start of the manuscript PDF file)?

Reviewer #1: Yes

Reviewer #2: Yes

5. Is the manuscript presented in an intelligible fashion and written in standard English?

Reviewer #1: Yes

Reviewer #2: Yes

6. Review Comments to the Author

Reviewer #1: The authors have addressed all comments.

Reviewer #2: In this study, the authors conducted Global trends in antimicrobial use in food-producing animals: 2020 to 2030 using estimate of usage in food-producing animals for 229 countries/territories in 2020. After adequate review, my comment is that this manuscript should be accepted in the present form.

7. PLOS authors have the option to publish the peer review history of their article (what does this mean?). If published, this will include your full peer review and any attached files.

**Do you want your identity to be public for this peer review?** For information about this choice, including consent withdrawal, please see our Privacy Policy.

Reviewer #1: No

Reviewer #2: **Yes: **Nma Bida Alhaji
